# Effect of Solvent and Monomer Ratio on the Properties of Polyphenylene Sulphone

**DOI:** 10.3390/polym15102279

**Published:** 2023-05-12

**Authors:** Azamat Zhansitov, Zhanna Kurdanova, Kamila Shakhmurzova, Azamat Slonov, Ilya Borisov, Svetlana Khashirova

**Affiliations:** Collective usage center “Polymers and Composites”, Kabardino-Balkarian State University Named after H.M. Berbekov, St. Chernyshevsky, 173, 360004 Nalchik, Russia

**Keywords:** polyphenylene sulfone, synthesis, solvent, monomer ratio, molecular mass, heat resistance, thermostability, membranes

## Abstract

For the first time, the effect of the solvent and monomer ratio on molecular weight, chemical structure, and mechanical, thermal, and rheological characteristics of polyphenylene sulfone has been studied. When dimethylsulfoxide (DMSO) is used as a solvent, cross-linking occurs during the processing of the polymer, which is accompanied by an increase in melt viscosity. This fact sets a pressing need for the complete removal of DMSO from the polymer. The best solvent used for the production of PPSU is N,N-dimethylacetamide. This study of the molecular weight characteristics of polymers by gel permeation chromatography showed the stability of the polymers practically does not change with a decrease in molecular weight. The synthesized polymers correspond in tensile modulus to the commercial analog Ultrason-P, while exceeding it in terms of tensile strength and relative elongation at break. Thus, the developed polymers are promising for spinning hollow fiber membranes with a thin selective layer.

## 1. Introduction

Aromatic polysulfones are a group of heat-resistant polyarylenes that remain functional at temperatures from −100 to +250 °C [1]. There are three main commercially available types of polysulfones: polysulfone (PSU), polyphenylene sulfone (PPSU), and polyethersulfone (PES). As amorphous thermoplastics, PSU, PESU, and PPSU offer, with their glass transition, temperatures of 190, 225, and 220 °C, respectively.

From our point of view, PPSU deserves special attention, since it is of great interest to various industries due to its particular set of valuable properties, such as heat resistance, fire resistance, good mechanical and dielectric properties, chemical and radiation resistance, etc. [2]. PPSUs are widely used in the automotive, medicine, electrical engineering, and aircraft industries and, in addition to traditional processing methods, are widely used in 3D printing [3,4,5]. PPSU is a perspective polymer material for the production of filtration membranes with unique properties: high long-term thermal stability, high mechanical strength, and increased resistance to hydrolysis, plasticization, and cracking under the action of a number of organic solvents compared to other polymers [6,7,8,9].

One of the most important characteristics that predetermine the possibility of manufacturing products from PPSU by injection molding is the polymer melt flow rate, which directly depends on the Mw. In this regard, the main industrial grades of PPSU (Ultrason^®^ P from BASF and Radel R from Solvay) have a fixed MFI value that is optimal for processing by injection molding. At the same time, the vast majority of research on the use of PPSU, for example, in 3D printing [5] or for the manufacture of membranes [9], is limited, first of all, by the rheological characteristics of these grades.

In this regard, it is quite important to determine methods for the targeted regulation of molecular weight characteristics. This is necessary to produce polymers with the required properties while maintaining the technological properties.

An important parameter in polymer synthesis, especially on an industrial scale, is the reproducibility of the Mw values from synthesis to synthesis. Modern synthesis methods, which allow for the obtaining of high molecular compounds with desired properties and characteristics, are largely based on the rule of nonequivalence of functional groups [4]. The stoppage of polymer chain growth during nonequilibrium polycondensation with an excess of one of the monomers is due to the fact that functional groups of the second type are absent at a certain stage of the reaction. The resulting macromolecules will have the same functional groups of the excess component at both ends of the chain, excluding further elementary reaction acts that lead to the growth of the polymer chain [9]. The proposed synthesis method makes it possible to produce polymers with the required end groups.

It is known [10,11,12] that the presence of labile terminal hydroxyl groups in polyethersulfones initiates thermal oxidation processes. At the same time, blocking the terminal hydroxyl groups, for example, with an excess of a dihalogen-containing monomer, increases the thermal stability and manufacturability of the resulting polymers by preventing structuring processes.

Currently, the main method for the synthesis of aromatic polyethersulfones is the method of high-temperature polycondensation by the mechanism of nucleophilic substitution of dihaloarylene sulfones with bisphenols in aprotic dipolar solvents in the presence of alkali metal carbonates. Studies in the field of bisphenol and dihaloaromatic compounds, as well as the study of the solvating ability of phenoxide ions (Na^+^ and K^+^) in dipolar aprotic dipolar solvents, have been studied in detail [13].

Bisphenols containing electron acceptor bridging groups (–SO_2_–; –SO–; –CO–) are less reactive than monomers containing electron donor bridging groups (–C(CH_3_)_2_–; –S–; –O–) due to a decrease in the negative charge on the oxygen atom [14]. The use of an aqueous solution of alkali metal hydroxide in the synthesis of polyesters has not found wide industrial application due to some disadvantages: the need for preparation of a concentrated aqueous solution followed by filtration and titration; the need for accurate dosing of the base; the need for a strong base that can lead to hydrolysis of the aromatic dihalide, which changes the stoichiometry of the starting monomers; and the need to carry out the synthesis in several stages [14]. To address these shortcomings, an approach has been developed for the production of polyethersulfones using potassium carbonate to synthesize a high molecular weight polymer. An additional advantage is the use of an excess of potassium carbonate, which is not extremely critical to achieve a high molecular weight [15,16,17].

The reactivity of dihaloaromatic monomers depends not only on the structure of the bridging groups, but also on the type of halogen. The high activity of fluorine compared to chlorine in the reaction of nucleophilic substitution is due to its greater electronegativity [14]. However, fluorinated monomers are very expensive, which leads to an increase in the cost of the polymer by several times. This problem can be solved by introducing a sulfo group into the dihalogenide, which has a high electron acceptor. Thus, it is possible to apply dichloro derivatives for fast polymerization.

A major challenge in this type of polycondensation is the choice of a solvent suitable for all compounds involved in the reaction. The synthesis of polysulfones requires solvents in which the initial monomers, intermediate compounds, and the polymer are dissolved. Since aprotic dipolar solvent dissolves ions better than protic solvents, the use of the former leads to a growth in the nucleophilicity of phenolates, and as a result, the reaction rate constants increase. Suitable solvents for polycondensation of polysulfones are dimethylsulfoxide (DMSO), N,N-dimethylacetamide (DMAA), N-methylpyrrolidone (N-MP), N,N-dimethylformamide, diphenylsulfone, dimethylsulfone, and sulfolane [14]. Despite the availability of publications on the synthesis of polyaryl sulfones, there are practically no results from studying the effect of synthesis conditions on the physicochemical properties of polyphenylene sulfone.

The purpose of this study was to determine the influence of the ratio of monomers and solvent type on molecular weight, chemical structure, and mechanical, thermal, and rheological characteristics of polyphenylene sulfone.

## 2. Materials and Methods

### 2.1. Characterisation and Methods

The molecular weight characteristics of the synthesized polymers were calculated according to the standard procedure with respect to monodisperse polystyrene standards by gel permeation chromatography on a Waters system with a differential refractometer (Chromatopack Microgel-5 (Waters, Milford, MA, USA), chloroform eluent, flow rate 1 mL/min).

IR spectra were recorded on a Fourier spectrometer (Spectrum Two; PerkinElmer, Inc., Waltham, MA, USA) in the range of 4000–450 cm^−1^ with a spectral resolution of 0.4 cm^−1^.

The glass transition temperature of the synthesized polymers was determined by differential scanning calorimetry (DSC) on a DSC 4000 instrument (PerkinElmer, Inc., Waltham, MA, USA). The quantity of 5–10 mg of the sample was sealed in an aluminum hermetic pan and heated to 250 °C at 10 °C/min under a flow of nitrogen (20 mL/min).

Thermogravimetric analysis (TGA) of the synthesized polymers was carried out on a TGA 4000 instrument (PerkinElmer, Inc., Waltham, MA, USA) under a flow of air (20 mL/min). at a heating rate of 10 °C/min in the range from 30 to 750 °C.

The melt flow index (MFI) of polymers was determined at a temperature of 350 °C and a load of 5 kg on a capillary viscometer (PTR-LAB-02, LOIP, Saint Petersburg, Russia).

Mechanical tests were carried out on a universal testing machine, Gotech Testing Machine CT-TCS 2000 (Taichung, Taiwan), at 23 °C. Impact tests were performed with and without notch, by the Izod method on the instrument Gotech Testing Machine, Model GT-7045-MD (Taichung, Taiwan), with the energy of a pendulum, 11 J.

The cross-linking and thermal stability of PPSU were investigated using a Tsvet-800 gas chromatograph (Tsvet, Dzerzhinsk, Russia) with a thermal conductivity detector and an absorption column with a length of 5 m filled with polysorb-1, impregnated with a 5% solution of polyethyleneglycoladipinate. The content of H_2_ was analyzed under the following conditions of the chromatographic procedure: volume flow of the argon carrier gas of 30 mL/min; detector temperature of 160 °C; evaporator temperature of 100 °C; column thermostat temperature of 100 °C; and detector current of 80 mA.

### 2.2. Chemicals

4,4’-Dihydroxy diphenyl and 4,4′-Dichlorodiphenyl sulfone was kindly supplied by Alfa Aesar (Heysham, UK) (99%). Potassium carbonate was purchased from Reachem (Moscow, Russia). N,N-dimethylacetamide (DMAA) and N-methylpyrrolidone (N-MP) were purchased from EKOS-1 (Moscow, Russia) (99%), and dimethyl sulfoxide (DMSO) and toluene were purchased from Vecton (Saint Petersburg, Russia) (99%).

### 2.3. Synthesis of PPSU

Polyphenylene sulfone (PPSU) synthesis was carried out in a 500 mL three-neck flask equipped with a nitrogen inlet, a mechanical stirrer, a Dean-Stark trap, and a reflux condenser; 4,4′-dihydroxy diphenyl (55.86 g, 0.3 mol), 4,4′-dichlorodiphenyl sulfone (88.73 g, 0.306–0.345 mol), and potassium carbonate (51.82 g, 0.375 mol) were charged in a flask. Then, N,N-dimethyl acetamide (450 mL) as reaction solvent was added. The reaction mixture was gradually heated to 165 °C for 4 h to distill the dimethylacetamide–water mixture. After the temperature reached 165 °C, the reaction mixture was allowed to proceed at this temperature for 6 h. After synthesis, the mixture was discharged, and the formed salts were filtered. The reaction solution was slowly poured into the water, acidified by oxalic acid. The precipitated polymer was filtered and washed 10 times with hot distilled water and dried in a vacuum oven at 160 °C for about 12 h.

The temperature regime for the synthesis of PPSU depends on the choice of an aprotic dipolar solvent. In the case of using DMAA and NMP, the synthesis temperature corresponds to the boiling point of the solvent of 165 and 202 °C, respectively, while the solvent forms an azeotropic mixture with water, and there is no need to use an inert nonpolar solvent (toluene, chlorobenzene, benzene, etc.). Synthesis using DMSO is carried out at a temperature of 160 °C, and an azeotropic former is always used.

The resulting polymer solution was filtered from the salts formed during the synthesis and precipitated by spraying into distilled water. The polymer was washed many times with hot distilled water and dried at 150 °C in a vacuum oven for about 12 h.

In this work, the Mw of synthesized polymers was controlled by varying the molar ratio of 4,4′-dihydroxybiphenyl and 4,4′-dichlorodiphenylsulfone monomers under the same temperature–time synthesis regime.

## 3. Results and Discussion

### 3.1. Investigation of the Influence of the Solvent on the Properties of PPSU

In order to study the effect of solvents on the properties of polyphenylene sulfone, syntheses were carried out in such aprotic dipolar solvents as N-methylpyrrolidone (N-MP), dimethyl sulfoxide (DMSO), and N,N-dimethylacetamide (DMAA). The choice of these solvents is because of their economic feasibility and ease of removing residual solvent from the synthesized polymer.

For a correct comparison of the properties of polymers synthesized in different solvents, samples with close intrinsic viscosities corresponding to commercial grades of PPSU were determined (Table 1).

Analysis of the IR spectra of polyphenylene sulfones synthesized in DMAA, DMSO, and MP demonstrates the absence of noticeable differences, which indicates the absence of the effect of these solvents on the structure of the resulting PPSU (Figure 1).

Absorption bands caused by stretching vibrations of C-H bonds of the aromatic ring are characterized by low intensity and appear as two peaks with maxima at 3037 and 3069 cm^−1^. Intense bands in the region of 1600–1400 cm^−1^, associated with skeletal vibrations of aromatic C=C bonds, appear as bands with maximums at 1584, 1486, and 1409 cm^−1^ [18].

The position of the band in the region below 900 cm^−1^ is determined by the presence and nature of substituents on the benzene ring. The type of substitution of the benzene ring is determined by the number and position of the bands in this region of the spectrum. For benzene, this vibration corresponds to a band at 671 cm^−1^. The introduction of substituents into the benzene ring leads to the appearance of longer wavelength frequencies [19].

As can be seen from Figure 1 in the PPSU spectrum, similar out-of-plane CH bending vibrations characteristic of 1,4-substituted benzene rings appear as a band with maxima at 869 and 829 cm^−1^. Planar bending vibrations of CH of p-substituted benzene rings appear as bands with maxima at 1105, 1073, and 1007 cm^−1^.

Polyarylsulfones also have very characteristic absorption in the region of 1350–1300 cm^−1^ and 1170–1120 cm^−1^, caused by antisymmetric and symmetric vibrations of the SO_2_ group, respectively. These bands are very intense and easily identifiable [19]. In the spectra of the synthesized polyphenylene sulfones, similar antisymmetric and symmetric vibrations of the SO_2_ group appear as a split band with maxima in the region of 1322 and 1294 cm^−1^, as well as a second split band with maxima in the region of 1165 and 1149 cm^−1^, respectively.

Vibrations of the C-O-C bond appear in the IR spectra as an intense absorption band in the region of 1200–1000 cm^−1^. The position of this band depends on the structure of the ester: for alicyclic esters, it lies in the region of 1150–1060 cm^−1^; for aromatic and unsaturated esters, it lies in the region of 1270–1200 cm^−1^. In the synthesized polyphenylene sulfones, asymmetric stretching vibrations of the C-O-C group appear in the region of 1235 cm^−1,^ which corresponds to the structure of aromatic esters.

Despite the fact that according to IR spectroscopy, polymers synthesized in different solvents have the same chemical structure, it is worth noting that samples synthesized in DMSO and in N-MP turned out to be darkened (Figure 2) after the molding. In this case, the samples synthesized in DMAA were light-colored. It is noteworthy that the polymer powders were not dyed before processing. Furthermore, the sample synthesized in DMSO has significantly lower MFI values (Table 2) compared to the polymers synthesized in NMP and DMAA, despite the fact that the intrinsic viscosities of these polymers are close. The reasons for this phenomenon are apparently associated with chemical processes occurring at a processing temperature of up to 350 °C.

It is known [11,20,21] that at 300–425 °C, the terminal phenoxy groups of polyethersulfones are attached to the phenylene fragments of the polymer chain, leading to the elimination of a hydrogen atom from the phenyl ring, which facilitates its cross-linking according to Figure 1:

However, the influence of trace amounts of solvent on cross-linking processes cannot be ruled out. Therefore, the content of trace amounts (0.05 wt%) of DMSO in the polymer leads to an increase in the melt flow index and a significant darkening of the polymer [22].

To elucidate the effect of the solvent on the cross-linking processes of polyphenylene sulfones at elevated temperatures, the volatile degradation products of PPSF were analyzed by gas chromatography. For this, a special cell was developed, the design of which is described in [23]. The degree of cross-linking of the polymers was evaluated by the amount of H_2_ released during the heating of polyphenylene sulfones in the temperature range of 250 to 450 °C.

As can be seen in Figure 3, when polymers are heated to 300 °C, the amount of hydrogen released is approximately the same for all samples.

However, at higher temperatures for the PPSU synthesized in DMAA, the H_2_ yield is 1.5–2 times lower than in the case of the other two solvents. According to the TGA data, the weight loss start temperature of PPSU is above 450 °C. Therefore, at lower temperatures, the release of gaseous hydrogen cannot be associated with the destruction of polymer chains. The reason for hydrogen formation is related to polymer cross-linking processes [23]. This is confirmed by a noticeable increase in the amount of hydrogen released at temperatures above 450 °C for all PPSU samples (Figure 3).

The effect of the solvent on the thermal-oxidative stability of PPSU samples was also revealed in this study of the stability of polymer melts by holding them in a capillary viscometer (Table 2). The MFI of the PPSU samples was measured after 5 and 20 min of exposure at a temperature of 350 °C. In this case, the loading of the polymer was 5 kg.

Table 2 shows that the sample synthesized in DMA has the most stable melt properties. The flow index of the sample synthesized in NMP increases by a slightly larger value from 23.0 to 23.4 g min^−1^, but the difference is not significant. A dramatic change in melt viscosity is observed for the polymer synthesized in DMSO. For this sample, the MFI increases by more than four times. This fact confirms our assumption that the processes of cross-linking of the polymer occur. This fact sets high requirements for the completeness of the removal of DMSO from the polymer.

A set of requirements is imposed on the synthesized polymer, including the required molecular weight, rheological and operational properties, and low costs for purification from impurities. Of all the solvents studied, DMA is most easily removed from PPSU. Since it forms an azeotrope with water, there is no need to use inert nonpolar solvents (toluene, chlorobenzene, and benzene) to remove water from the reaction medium.

Thus, the ease of removing the residual DMA solvent from the PPSU leads to the production of powders that form melts with stable properties and are easily processed into a final product with desired properties. In addition, such products are more attractive from an aesthetic point of view, since they are less intensely colored. Hence, the best solvent used for the production of PPSU is N,N-dimethylacetamide.

### 3.2. Investigation of the Influence of the Ratio of Components on the Properties of Polyphenylene Sulfone

In this work, the Mw of polymers was controlled by varying the molar ratio of 4,4′-dihydroxybiphenyl and 4,4′-dichlorodiphenylsulfone (DCDPS) monomers within 1:1–1.15 under the same temperature–time synthesis regime. Using this method, we synthesized polymers with a molecular weight from 13,400 to 102,000 g/mole.

This study of the molecular weight characteristics of polymers by gel permeation chromatography showed that the synthesis of PPSU with blocking of the end groups led to the production of polymers with a unimodal molecular weight distribution. At the same time, with an increase in the excess of DCDPS, a regular decrease in the molecular weight and an increase in the MFI of the synthesized polymers are observed (Figure 4 and Table 3).

An analysis of the IR spectra of the obtained polymers showed that in the case of using 4,4’-dichlorodiphenylsulfone as a blocker of end groups, characteristic peaks appear in the IR spectra in the region of 757 cm^−1^ (Figure 5a) and 473 cm^−1^ (Figure 5b) corresponding to Ph-Cl bonds [19], while these peaks are absent in the hydroxyl-terminated polymer synthesized with an excess of 4,4’-dihydroxydiphenyl.

At the same time, with an increase in the amount of excess 4,4’-dichlorodiphenylsulfone, the intensity of these peaks naturally increases. From the intensity of these peaks, one can indirectly draw conclusions about the molecular weight of PPSU synthesized with blocked Cl end groups.

Table 4 shows that the thermal stability of the synthesized polymers practically does not change in the studied range of molecular weights. Their destruction in air proceeds in two stages. The first stage, in the range of 470–550 °C, is due to chain-breaking reactions and the loss of volatile components, and the second stage is due to thermal oxidation. The second stage of decomposition proceeds at lower rates and ends in the range of 740–770 °C without the formation of coke residue.

As the molecular weight increases up to Mw = 40,000 g/mol, the glass transition temperature rises sharply. With a further increase in the molecular weight, the growth in the glass transition temperature slows down and reaches the limiting values (Figure 6). The glass transition temperature is one of the most important properties of a polymer, primarily determining its heat resistance. Polymers with a high glass transition temperature are preferred. However, a compromise has to be found between the heat resistance and viscosity of the polymer composition. With an increase in the molecular weight of the polymer, the melt reaches the viscosity limit, at which it is not possible to process it by extrusion.

The influence of the molecular weight of the polymer on its suitability for processing and the mechanical properties of samples cast on the basis of polymers with different molecular weights was investigated. The results of this study of the physical and mechanical properties of the synthesized PPSU produced by injection molding are shown in Table 5.

We tried to make samples for mechanical study from polymers of all available molecular weights. However, the sample produced from polymer with a 2% excess (PPSU-1) monomer was found to be unsuitable for injection molding due to the high viscosity of the melt. PPSU synthesized with an excess of DCDPS equal to 7% (PPSU-4); on the contrary, having a melt viscosity that was too low led to difficulties in the molding of samples. These samples, in turn, were characterized by high brittleness and were destroyed immediately upon removal from the mold. The lowest molecular weight sample, obtained with the maximum excess (15%, (PPSU-5)), turned out to be unsuitable for the molding. PPSU-2 and PPSU-3 samples with MFI values 4–21 (g min^−1^) corresponding to injection grades of polymeric materials were successfully molded. Therefore, there is a rather narrow range of molecular weights 40–60 × 10^−3^ (g/mole)) in which the polymer is suitable for processing by extrusion. This study of the mechanical properties of the PPSU samples shows that with a decrease in Mw, the brittleness of the material naturally increases, which is expressed in a drop in impact strength and relative elongation at break. Additionally, with an increase in the amount of excess monomer and, accordingly, a decrease in Mw, there is a tendency for a slight increase in the elastic modulus. Note that the synthesized polymers correspond to the commercial analog Ultrason-P in terms of tensile modulus but surpass it in terms of tensile strength and elongation at break.

## 4. Conclusions

For the first time, the influence of the solvent and the ratio of monomers on the molecular weight, chemical structure, and mechanical, thermal, and rheological characteristics of polyphenylene sulfone was studied. Samples synthesized in DMSO and in N-MP turned out to be darkened after the molding. PPSU samples synthesized in DMAA were light-colored. Moreover, the sample synthesized in DMSO has significantly lower MFI values compared to those of polymers synthesized in NMP and DMAA despite the fact that the intrinsic viscosities of these polymers are close. The reasons for this phenomenon are apparently associated with chemical cross-linking processes occurring at a temperature of up to 350 °C. It was found that the most suitable solvent for the production of PPSU is N,N-dimethylacetamide. The synthesized polymer has the necessary set of molecular weight, rheological, and physico-mechanical properties with the least laboriousness of the process.

The dependence of the molecular weight of the synthesized PPSU on the ratio of monomers was determined, which makes it possible to obtain polymer samples with a fixed molecular weight for various applications. The dependence of the thermal and mechanical properties of PPSU on Mw was studied. It is shown that with a decrease in Mw, the glass transition temperature logically decreases, and the brittleness of the material increases, which is expressed in a decrease in impact strength and relative elongation at break. It was found that there is a rather narrow range of PPSU molecular weights 40–60 × 10^−3^ (g/mole)) in which the polymer is suitable for processing by extrusion. Synthesized polymers correspond to the commercial analog Ultrason-P in terms of tensile modulus, but surpass it in terms of tensile strength and elongation at break.

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
