# Peer review of "Effect of Solvent and Monomer Ratio on the Properties of Polyphenylene Sulphone"

_polymers, 2023, doi:10.3390/polym15102279_

Round 1
Reviewer 1 Report
The article by Zhansitov et al., titled "Effect of Solvent and Monomers Ratio on the Properties of Polyphenylene Sulphone," examines the impact of solvent and monomer ratio on the properties of polyphenylene sulfone (PPSU). The study investigates several characteristics of PPSU, including molecular weight, chemical structure, mechanical, thermal, and rheological properties. It was found that using DMSO as a solvent led to cross-linking during processing, resulting in increased melt viscosity. Complete removal of DMSO from the polymer was necessary, and N,N-dimethylacetamide was identified as the best solvent for PPSU production.
The manuscript has many flaws. I suggest that the authors revise the paper thoroughly and then re-submit it. In its present form, I do not consider the manuscript publishable.
1. General comment 1:
The manuscript of Zhansitov et al. requires significant improvements in language and punctuation, particularly with regard to scientific precision. The paper contains major errors that affect the clarity and accuracy of the findings, which could have an impact on the scientific community's ability to replicate and build upon the research. It is crucial that the authors revise and improve the language and punctuation of their manuscript to ensure that the findings are presented clearly and accurately.
2. General comment 2:
The explanations of your observations in the results and discussion appear to be lacking in depth, and the conclusions drawn are insufficient. Such a style does not meet the expectations of good scientific writing. As a result, I would highly recommend that you provide a detailed and comprehensive discussion and interpretation of the results for each section. This is crucial for readers to fully understand the significance and implications of your findings. By doing so, you will ensure that your research is scientifically sound and credible.
3. General comment 2:
The Conclusion section of the paper appears to be lacking in clarity and focus. Rather than providing a concise summary of the main findings and implications of the study, it reads more like a general discussion that was missing from the preceding chapters. This weakens the overall impact of the research and may lead to confusion for readers. As a result, I strongly recommend that the authors re-write this section with a greater emphasis on the main output of their research and a clear and concise summary of their key findings. This will help to ensure that the study's contribution to the field is clearly understood and appreciated by readers.
Here some additional comments:
4. The first part of the introduction Line 1-44 is too long and generic. Please make it more concise and focus on the key message of the paper.
5. Line 93, Materials and Methods: All chemicals used, their purities and suppliers must be mentioned.
6. Line 119, Synthesis of PPSU: The quantities and molar equivalents of all chemicals used are missing. Also, reaction times, quantities of washing solutions etc. are missing. The synthesis cannot be reproduced as it is now and must not be published in its present form!
7. Line 140: I don’t see any reason to show the mechanism of the synthesis here. It is a simple condensation, that we teach students in the first year. Bring it in the introduction or delete it. There is no reason to show it in section 3.
8. Line 153, IR-spectra: highlight the representative numeric peak values in the spectrum.
9. Line 158, text IR-spectra: You explain your observations, but don’t draw any conclusions. This is not good scientific style. I ask to show a sufficient discussion and interpretation of the results for each section.
10. Line 169: You do mention the viscosities, but do not compare them. Are they good? Do you expect those values? How are they related to similar polymers in this field? Pleas add a discussion.
11. Line 179: Which “various” chemical reactions. What is your hypothesis?
see general comment 1:
The manuscript of Zhansitov et al. requires significant improvements in language and punctuation, particularly with regard to scientific precision. The paper contains major errors that affect the clarity and accuracy of the findings, which could have an impact on the scientific community's ability to replicate and build upon the research. It is crucial that the authors revise and improve the language and punctuation of their manuscript to ensure that the findings are presented clearly and accurately.
Author Response
- General comment 1:
The article has been revised from the point of view of the language.
- General comment 2:
The discussion of the results was expanded in accordance with the requirements of the reviewer.
- General comment 3:
The conclusions have been rewritten taking into account the comments.
- The conclusions have been rewritten taking into account the comments.
- Added information about brands and suppliers of used reagents.
- Changes have been made to the section "Synthesis of PPSU", information on the amounts of reagents used has been added;
- The synthesis scheme has been deleted from the discussion of the results section.
- On the IR spectra, the numerical values of the main peaks are highlighted and a description is added.
- In line 158, the description of the IR spectra has been revised in accordance with the recommendations of the reviewer.
- The viscosity of the polymers taken for comparison (Table 1) corresponds to the industrially produced injection grade PPSU Ultrason P.
- In line 179, the not sufficiently precise phrase "various chemical reactions" was replaced with the clarifying "chemical reactions of polymer crosslinking"
Thanks for the comments. Thanks to your work, the article has improved significantly.
Reviewer 2 Report
The authors presented a thorough study on the effect of solvent and monomer ratio on PPSU properties. The motivation and the potential impact of this study were clearly presented as well. I would suggest the minor revision of this work before being accepted.
1. Please specify suppliers, and grades for different chemicals used in this study.
2. Please include DSC curves in this work (either in main context or supporting information).
Author Response
1. Added information about brands and suppliers of used reagents.
2. In the Supplementary file, DSC dependencies of the synthesized polymers are added.
Round 2
Reviewer 1 Report
Corrections were made according to the suggestions. However, the overall presentation, especially the figures, should be improved before publication.
There are still a lot of errors and inaccuracies. I recommend a language correction service!